# Power-Law between the Apparent Drainage Density and the Pruning Area

Soohyun Yang[1,2], Kwanghun Choi[2], and Kyungrock Paik[2]

[1] Department of Civil and Environmental Engineering, Seoul National University, 1 Gwanak-ro, Gwanak-gu, Seoul, 08826, South Korea (current affiliation)
[2] School of Civil, Environmental, and Architectural Engineering, Korea University, 145 Anam-ro, Seongbuk-gu, Seoul, 02841, South Korea

*Correspondence to*: Kyungrock Paik (paik@korea.ac.kr)

**Abstract.** Self-similar structures of river networks have been quantified as diverse scaling laws. Among them we investigated a power functional relationship between the apparent drainage density $\rho_a$ and the pruning area $A_p$ with an exponent $\eta$. We analytically derived the relationship between $\eta$ and other scaling exponents known for fractal river networks. The analysis of 14 real river networks covering diverse range of climate conditions and free-flow connectivity levels supports our derivation. We further linked $\eta$ with non-integer fractal dimensions found for river networks. Synthesis of our findings through the lens of fractal dimensions provides an insight that the exponent $\eta$ has fundamental roots in fractal dimension for the whole river network organization.

## 1 Introduction

Since first proposed by Horton (1945), the drainage density $\rho$ has long been recognized as an important metric to describe geomorphological and hydrological characteristics of a catchment. Defined as $\rho = L_T / A_T$ where $A_T$ is the total catchment area, $\rho$ is a function of the total channel length $L_T$ in a catchment. Alternatively, $\rho$ is a function of the channel forming area $A_o$ (also called the source area or the critical contributing area) (Band, 1986; Montgomery and Dietrich, 1988; Tarboton et al., 1988), which is directly related to $L_T$. The spatial variation of $\rho$ among catchments is associated with their climates (Melton, 1957; Madduma Bandara, 1974; Wang and Wu, 2013), which can be represented by measures such as the precipitation effectiveness (PE) index (Thornthwaite, 1931). Also over time, $A_o$ and so $\rho$ of a given catchment dynamically vary. $A_o$ reduces as the catchment becomes wetter, water accumulates more readily in the soils of low-gradient areas, and saturated areas expand accordingly. This mechanism leads to the enlargement of the stream network (greater $L_T$). Conversely, when the catchment gets drier, $A_o$ increases, which in turn results in the contraction of the stream network (Godsey and Kirchner, 2014; Hooshyar et al., 2015; Durighetto et al., 2020). Therefore, $L_T$ and $\rho$ are inversely related to $A_o$ (Tarboton et al., 1991).

On another note, the 'rate' at which $L_T$ (and so $\rho$) varies with $A_o$ is likely determined by the shape of landscape or a given topography. The close relationship between the main channel length $L$ and the drainage area $A$ is well known as a power function with a positive exponent $h$ (Hack, 1957), i.e.,

$$L \propto A^h, \tag{1}$$

which provides a clue about the relationship between $L_T$ and $A_o$. However, they differ in two senses: (1) $L_T$ is the total length counting all tributaries, while $L$ is the length of the main channel only; and (2) $L$ is the length within the area $A$ while $L_T$ is the length of channels excluded from $A_o$. $L_T$ reduces as $A_o$ increases, while $L$ grows with $A$ (Eq. (1)).

The usage of the digital elevation models (DEMs) in the river network analysis introduced a constant called the pruning area $A_p$. In extracting a stream network from a DEM, cells of the upslope area $A$ less than $A_p$ are considered as hillslope and excluded from the network. For the ideal delineation of a river network, $A_p$ is expected to be $A_o$. However, $A_p$ is an any arbitrary value

and differs from $A_o$ by definition. If $A_p = 0$, every DEM cell is considered as channel while $A_p$ can be as large as $A_T$ for a completely dry landscape. As $A_p$ increases, less channels are extracted, resulting in a smaller 'apparent' drainage density $\rho_a$.

We distinguish $\rho_a$ from the real drainage density $\rho$, accommodating the difference between $A_p$ and $A_o$. It was found that $\rho_a$ decreases as $A_p$ grows following a power function (Moglen et al., 1998), i.e.,

$$\rho_a \propto A_p^{-\eta}. \tag{2}$$

While Eq. (2) should be distinguished from the relationship between $\rho$ and $A_o$, it reflects the topographic characteristic which is likely similar for the relationship between $\rho$ and $A_o$.

The background described above naturally leads us to the basic question about the physical origin of the power-law Eq. (2) and its scaling exponent $\eta$. $L_T$ has been expressed as a power function of the discharge at the catchment outlet $Q$ (Godsey and Kirchner, 2014; Hooshyar et al., 2015; Jensen et al., 2017), i.e., $L_T \propto Q^\beta$. Prancevic and Kirchner (2019) derived the exponent $\beta$ as the combination of $\eta$ and two other scaling exponents found in topographic attributes, i.e., $\beta = \eta / (\theta + \gamma + 1)$, where $\theta$ is the power-law exponent relating local channel slope to drainage area called the concavity (Montgomery and Foufoula-

Georgiou, 1993; McNamara et al., 2006), and $\gamma$ is the exponent of a hypothetical power function between $A$ and valley transmissivity $T$ (the product of subsurface cross-sectional area and conductivity, which in turn is expressed in units of cubic length per time (Prancevic and Kirchner, 2019). Adopting this, we can reason $\eta = \beta(\theta + \gamma + 1)$. However, Prancevic and Kirchner (2019) acknowledged that the above expression of $\beta$ is yet to be generalized across a range of sizes and landscapes. Eq. (2) and the exponent $\eta$ have awaited for deeper investigations.

Moglen et al. (1998) attempted direct DEM analyses to investigate the $\rho_a$–$A_p$ relationships in real river networks. But, $A_o$ and $A_p$ were undistinguished and little discussion about $\eta$ was given. Further, topographic data they adopted were limited, while a greater resolution DEM for catchments of known $A_o$ or blue-lines are needed to properly approach the given subject with terrain analyses. It is worth to realize that the power-law relationship of Eq. (2) implies fractal network formation. A river network is fractal, and many regular power-laws have been reported as characteristic signatures of a naturally evolved river network

(Dodds and Rothman, 2000). As the power-law relationship between $\rho_a$ and $A_p$ can also serve as a signature reflecting the self-similarity, it is plausible to claim the linkage between $\rho_a$–$A_p$ relationship and other power-laws known in natural river networks.

The exponent $\eta$ brings further interesting questions. In Eq. (2), $\eta = 0.5$ is anticipated to satisfy dimensional consistency (Tarboton et al., 1991). But the rough analysis of Moglen et al. (1998) raises a doubt whether $\eta$ estimated from any real catchment meets this consistency. This issue is analogous to the question about the exponent $h$ in Eq. (1), which should also be 0.5 to keep

consistency in dimension (Hjelmfelt, 1988). In fact, $h$ values reported for natural rivers are mostly greater than 0.5, i.e., between 0.5 and 0.7 (Hack, 1957; Gray, 1961; Robert and Roy, 1990; Crave and Davy, 1997). This has brought the introduction of the fractal dimension (Mandelbrot, 1977), whose values for river networks range between 1 and 2 (e.g., Feder, 1988) (further detailed explanations are provided in Sect. 4). Similarly, we can claim that the dimensional inconsistency in Eq. (2), if any, can be resolved by incorporating the fractal dimension. It is also an open question what controls $\eta$. While the relationship between $\rho_a$

and $A_p$ reflects the topography, if $\eta$ is a fixed constant of 0.5, despite dimensional consistency, it implies limited role of topographic variation on $\eta$. If $\eta$ is variable, the underpinning mechanism that changes local catchment topography and so $\eta$ is to be explored. In particular, we are curious about roles of human intervention and ecosystem evolution in conjunction with climate forcing on the relationship between $\rho_a$ and $A_p$. To understand this, it is desired to investigate a range of catchments under different developmental stages and climate conditions as well.

Here, we aimed to corroborate the aforementioned claims and hypothesis about the $\rho_a$–$A_p$ relationship and its exponent $\eta$. To this end, in the next Sect. 2, we reviewed the scaling relationships known in a river network. Then, we presented analytical

derivation of Eq. (2), and demonstrated how this is related with other power-laws known for a river network. To support our argument, many real catchments under the wide range of climatic conditions and free-flow connectivity levels were analyzed with terrain analysis methods in a thorough manner using high resolution DEMs and trust-worthy blueline data. These are
described in Sect. 3. With these results, we explored physical meanings embedded in the power-law relationship between $\rho_a$ and $A_p$ with the notion of fractal dimension in Sect. 4. Summary and conclusions are given in Sect. 5.

## 2 Cross-Relationships among Scaling Laws

### 2.1 Review on the scaling laws of a river network

The river network has been perceived as an archetypal fractal network in nature (Mandelbrot, 1977; Rodríguez-Iturbe and
Rinaldo, 2001), exhibiting scale-invariant organization. Systematic measures for characterizing structural hierarchy help manifest the self-similarity. Horton-Strahler ordering scheme (Horton, 1945; Strahler, 1957) has been popularly employed to investigate their structural characters. In this framework, the number, the mean length, and the mean drainage area of $\omega$-order streams in a catchment, stated as $N_\omega$, $\bar{L}_\omega$, and $\bar{A}_\omega$, respectively, are defined for an order $\omega$ ranging from 1 to $\Omega$, where $\Omega$ is the highest order in the network. There is only one $\Omega$–order stream in a river network (i.e., $N_\Omega=1$). Then, the total channel length
$L_T$ used for the definition of the drainage density $\rho$, is given as

$$L_T = \sum_{\omega=1}^{\Omega} N_\omega \bar{L}_\omega. \tag{3}$$

Following its definition, the length of any lower order stream is excluded in $\bar{L}_\omega$. Therefore, $\bar{L}_\Omega$ is neither the upslope length $L$ of a main channel, nor $L_T$. By contrast, $\bar{A}_\omega$ includes the drainage area of all upstream branches (of $\omega - 1$ and lower orders), e.g., $\bar{A}_\Omega$ is identical to $A_T$. To resolve the discrepant definitions of $\bar{L}_\omega$ and $\bar{A}_\omega$, the cumulative mean length $\Gamma_\omega$ was proposed
to match the definition of area (Broscoe, 1959) as

$$\Gamma_\omega = \sum_{k=1}^{\omega} \bar{L}_k \tag{4}$$

which is an order-discretized approximation of $L$. Alternatively, to match the definition of length, the eigenarea, also called the interbasin area (Strahler, 1964) or the contiguous area (Marani et al., 1991), was proposed as the area directly draining to the $\omega$–order stream (Beer and Borgas, 1993). The mean eigenarea $\bar{E}_\omega$ of $\omega$–order streams is
$$\bar{E}_\omega = \bar{A}_\omega - \bar{A}_{\omega-1}(N_{\omega-1}/ N_\omega). \tag{5}$$

The self-similar structure of a river network has been captured through the linear scaling of above quantities ($N_\omega$, $\bar{L}_\omega$, $\bar{A}_\omega$, and $\bar{E}_\omega$) with $\omega$ on a semi-log paper (Horton, 1945; Schumm, 1956; Yang and Paik, 2017) as

$$N_\omega = R_B{}^{\Omega-\omega}; \ \bar{L}_\omega = \bar{L}_\Omega R_L{}^{\omega-\Omega}; \ \bar{A}_\omega = \bar{A}_\Omega R_A{}^{\omega-\Omega}; \text{ and } \bar{E}_\omega = \bar{E}_\Omega R_E{}^{\omega-\Omega} \tag{6}$$

where $R_B$, $R_L$, $R_A$, and $R_E$ are the bifurcation, the length, the area, and the eigenarea ratios, respectively. These dimensionless
ratios are often called the Horton ratios as a group. They are related to each other (Morisawa, 1962; Rosso, 1984; Tarboton et al., 1990) and typically range as $3 < R_B < 5$, $1.5 < R_L < 3$, and $3 < R_A < 6$ (Smart, 1972), and $R_E \approx R_L$ (Yang and Paik, 2017).

In addition to Eq. (6), power functional relationships between geomorphologic variates have also been found and served as evidence of the scale-invariant river network structures. The Hack's law (Eq. (1)) is a classical principle in this line. Another interesting power-law relationship lies in the exceedance probability distributions of upstream area. Using a theoretical
aggregation model, Takayasu et al. (1988) showed that the exceedance probability distribution of injected mass in a tree network always follows a power-law. In fact, their model is equivalent to the random-walk model of Scheidegger (1967) devised to mimic a river network (Takayasu and Nishikawa, 1986). Replacing the mass (flow) in the aforementioned study with the drainage area (which is rational if rainfall is spatially uniform), it leads to the power-law exceedance probability distribution of 'drainage area.' From all DEM cells composing a catchment, one can calculate the probability distribution of the

upslope area $A$ of a cell, i.e., $P(A)$, which is minimal for $A = A_T$ (as only one cell at the outlet meets this case). It is found that the probability for a randomly designated point to have $A$ exceeding a reference value $\delta$ ($0 \leq \delta \leq A_T$) decreases with $\delta$ (Rodríguez-Iturbe et al., 1992a), following a power-law as

$$P(A \geq \delta) \propto \delta^{-\varepsilon} \tag{7}$$

where the exponent $\varepsilon$ is reported as between 0.40 and 0.46 for most river networks (Rodríguez-Iturbe et al., 1992a; Crave and Davy, 1997). Above two power-laws (Eqs. (1) and (7)) are related as $h + \varepsilon = 1$ (Maritan et al., 1996), which suggests a trade-off between the two exponents by balancing each other with their respective ranges to form the catchment boundary within a confined space.

Two classes of scaling relationships reviewed above, i.e., Horton's laws (Eq. (6)) and power-law relationships are linked as shown by La Barbera and Roth (1994), i.e.,

$$\varepsilon = 1 - h = \frac{\ln(R_B/R_L)}{\ln R_A}. \tag{8}$$

Two other expressions, comparable to Eq. (8), appear in literature. de Vries et al. (1994) derived $\varepsilon = 1 - \ln R_L/\ln R_B$, which is a special case of Eq. (8) where $R_B = R_A$. Empirical studies support that $R_B$ is indeed close to $R_A$ (Smart, 1972). For a 'topological' Hortonian tree where no constraint on stream length in a finite area is given, Veitzer et al. (2003) and Paik and Kumar (2007) showed that $\varepsilon = \ln R_B/\ln R_A - 1$. This is another special case of Eq. (8) where $R_L = R_A$, the assumption used in the analysis of 'topological' self-similar trees where only connections among nodes matter with no spatial constraint (Paik and Kumar, 2007).

## 2.2 Linkage to $\rho_a$–$A_p$ relationship

The inverse relationship between the pruning area $A_p$ and the resulting apparent drainage density $\rho_a$ can be found in the DEM analysis (**Fig. 1**). Below, we analytically derived their plausible relationship (Eq. (2)), using the scaling relationships reviewed above. Through this investigation, we importantly revealed $\eta = \varepsilon$, i.e., the scaling exponents in Eqs. (2) and (7) are identical. We arrived at the same conclusion from two different approaches, described below.

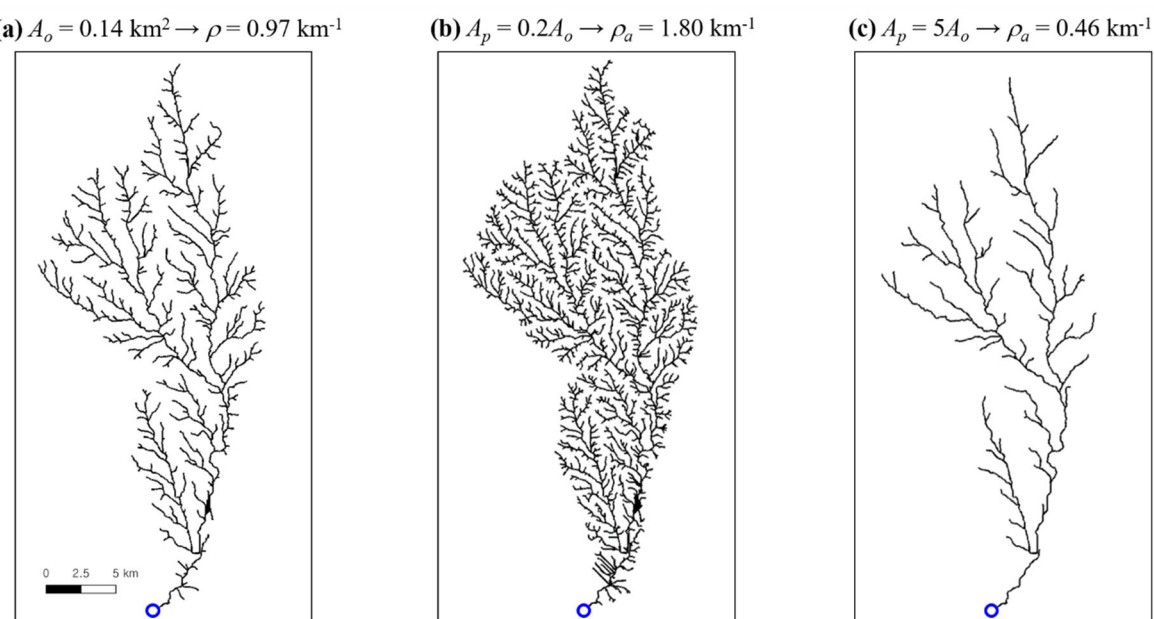

**(a)** $A_o = 0.14 \text{ km}^2 \rightarrow \rho = 0.97 \text{ km}^{-1}$      **(b)** $A_p = 0.2A_o \rightarrow \rho_a = 1.80 \text{ km}^{-1}$      **(c)** $A_p = 5A_o \rightarrow \rho_a = 0.46 \text{ km}^{-1}$

**Figure 1.** Stream network (black line) of the Brushy catchment, the USA, extracted with varying pruning area. A blue circle indicates the outlet. For this catchment, the channel forming area $A_o$ and the corresponding drainage density $\rho$ are available from the National Hydrography Dataset Plus Version 2 (NHDPlusV2) (McKay et al., 2012) (see Sect.3 for detail). (a) River network given in the NHDPlusV2, and the corresponding $A_o$ and $\rho$. (b – c) Extension and contraction of the river network with a pruning area $A_p$ that is 5 times smaller and greater than $A_o$, respectively. The apparent drainage density $\rho_a$ accordingly varies.

### 2.2.1 Derivation 1

For the Hortonian tree, $A_p$ can vary in a discrete manner (order-by-order), as we set $A_p = \bar{A}_\omega$. Given that up to $\omega$-order streams are pruned in a river network, the total length after pruning is $\sum_{k=\omega+1}^{\Omega} N_k \bar{L}_k$, by revising Eq. (3). Replacing $N_k$ and $\bar{L}_k$ in this equation with Eq. (6) leads to the expression of $\rho_a$ as

$$\rho_a = \frac{\bar{L}_\Omega}{\bar{A}_\Omega} \sum_{k=\omega+1}^{\Omega} R_B^{\,\Omega-k} R_L^{\,k-\Omega}. \tag{9}$$

The sum of above geometric series is

$$\rho_a = \frac{\bar{L}_\Omega}{\bar{A}_\Omega (R_B/R_L - 1)} \left[ \left(\frac{R_B}{R_L}\right)^{\Omega-\omega} - 1 \right]. \tag{10}$$

The logarithm of the term $(R_B/R_L)^{\Omega-\omega}$ in Eq. (10) can be written, using Eq. (6), as

$$\ln\left(\frac{R_B}{R_L}\right)^{\Omega-\omega} = (\Omega - \omega) \ln \frac{R_B}{R_L} = \frac{\ln(\bar{A}_\Omega/\bar{A}_\omega)}{\ln R_A} \ln \frac{R_B}{R_L} = \frac{\ln(R_B/R_L)}{\ln R_A} \ln \frac{\bar{A}_\Omega}{\bar{A}_\omega}. \tag{11}$$

Given that $\bar{A}_\omega = A_p$, from Eq. (11) we can state

$$(R_B/R_L)^{\Omega-\omega} = \left(\bar{A}_\Omega/A_p\right)^{\frac{\ln(R_B/R_L)}{\ln R_A}}. \tag{12}$$

Substituting this into Eq. (10) yields an approximate power-law, i.e.,

$$\rho_a = \frac{\bar{L}_\Omega}{\bar{A}_\Omega (R_B/R_L - 1)} \left[ \left(\frac{A_p}{\bar{A}_\Omega}\right)^{-\frac{\ln(R_B/R_L)}{\ln R_A}} - 1 \right] \propto A_p^{-\frac{\ln(R_B/R_L)}{\ln R_A}}. \tag{13}$$

Given that $R_B \approx R_A > R_L$ (Smart, 1972) for a typical river network, $-1 < -\ln(R_B/R_L)/\ln R_A < 0$. With this range and for $A_p \ll \bar{A}_\Omega$, $\left(A_p/\bar{A}_\Omega\right)^{-\ln(R_B/R_L)/\ln R_A} = \left(\bar{A}_\Omega/A_p\right)^{\ln(R_B/R_L)/\ln R_A} \gg 1$. This allows the approximation $[\left(\bar{A}_\Omega/A_p\right)^{\ln(R_B/R_L)/\ln R_A} - 1] \approx \left(\bar{A}_\Omega/A_p\right)^{\ln(R_B/R_L)/\ln R_A}$. Empirical studies suggested $A_o < 0.1\bar{A}_\Omega$ to characterize fluvial channel networks (Montgomery and Foufoula-Georgiou, 1993; McNamara et al., 2006), implying the scope of this derivation, i.e., $A_p \ll \bar{A}_\Omega$, of practical range. Comparing Eqs. (2) and (13), we can explicitly express

$$\eta = \frac{\ln(R_B/R_L)}{\ln R_A}. \tag{14}$$

This expression is identical to Eq. (8), which implies $\eta = \varepsilon$.

### 2.2.2 Derivation 2

The conclusion of $\eta = \varepsilon$ can also be derived by employing the eigenarea (Yang, 2016). Approximating an $\omega$-order sub-catchment as a rectangle, $\bar{E}_\omega$ can be rewritten as $\bar{E}_\omega = W\bar{L}_\omega$ where the mean overland flow length is $W/2$. As $W$ is regarded almost a constant (Hack, 1957; Yang and Paik, 2017), the apparent drainage density for the pruning area $A_p = \bar{A}_\omega$ becomes

$$\rho_a = \frac{1}{\bar{A}_\Omega} \sum_{k=\omega+1}^{\Omega} N_k \bar{L}_k = \frac{1}{\bar{A}_\Omega W} \sum_{k=\omega+1}^{\Omega} N_k \bar{E}_k. \tag{15}$$

On the other hand, $P(A \geq A_p)$ is defined from geometry as

$$P(A \geq A_p) = \frac{1}{\bar{A}_\Omega} \sum_{k=\omega+1}^{\Omega} N_k \bar{E}_k \tag{16}$$

which equals to $W\rho_a$ from Eq. (15). As $P(A \geq A_p) \propto A_p^{-\varepsilon}$ (Eq. (7)), we realize that $\rho_a \propto A_p^{-\varepsilon}$ and thereby $\eta = \varepsilon$. While equation (13) was derived for $A_p \ll \bar{A}_\Omega$, this alternative derivation shows the power-law regardless of the range in $A_p$. Earlier, we discussed the reciprocal nature of two relationships; one between $L_T$ and $A_o$, and the other between $L$ and $A$. Combining above conclusion of $\eta = \varepsilon$ and $h + \varepsilon = 1$, we realize that $\eta = 1 - h$, indeed implying the compensating function between them.

## 3 Analyses of Real River Networks

### 3.1 Data and methods

To evaluate the power-law Eq. (2) and the derivation of $\eta = \varepsilon$, we analyzed real river networks in the contiguous United States. We have chosen 14 study networks (**Fig. 2**) from the pool investigated in previous studies of Tarboton et al. (1991), Rodríguez-Iturbe et al. (1992a), Botter et al. (2007), Hosen et al. (2021), and Carraro and Altermatt (2022). They are carefully selected to cover distinct hydro-climatic regions and a range of free-flowing capacity (**Table 1**). The climate feature is described by the Köppen-Geiger climate classification (Beck et al., 2018). The free-flow characteristic is referred as an integrated connectivity status index (CSI) created at a global scale by Grill et al. (2019) for the first time. The CSI comprehensively and quantitatively describes the capacity of individual river reaches to freely flow based on the synthesis of observed and modelled datasets. The reported CSI values, ranging from 0 to 100 %, are the weighted average of estimated five pressure indicators – river fragmentation, flow regulation, sediment trapping, water consumption, and infrastructure development in riparian areas and floodplains – which represent natural and human inferences within longitudinal, lateral, vertical, and temporal dimensions. If a river reach loses connectivity due to any of aforementioned pressures, its CSI value decreases. We calculated a catchment-unit CSI by weighting the length of individual reaches in a given catchment. The CSI of our 14 catchments ranges from 58 to 100 % which is irrelevant to each catchment size.

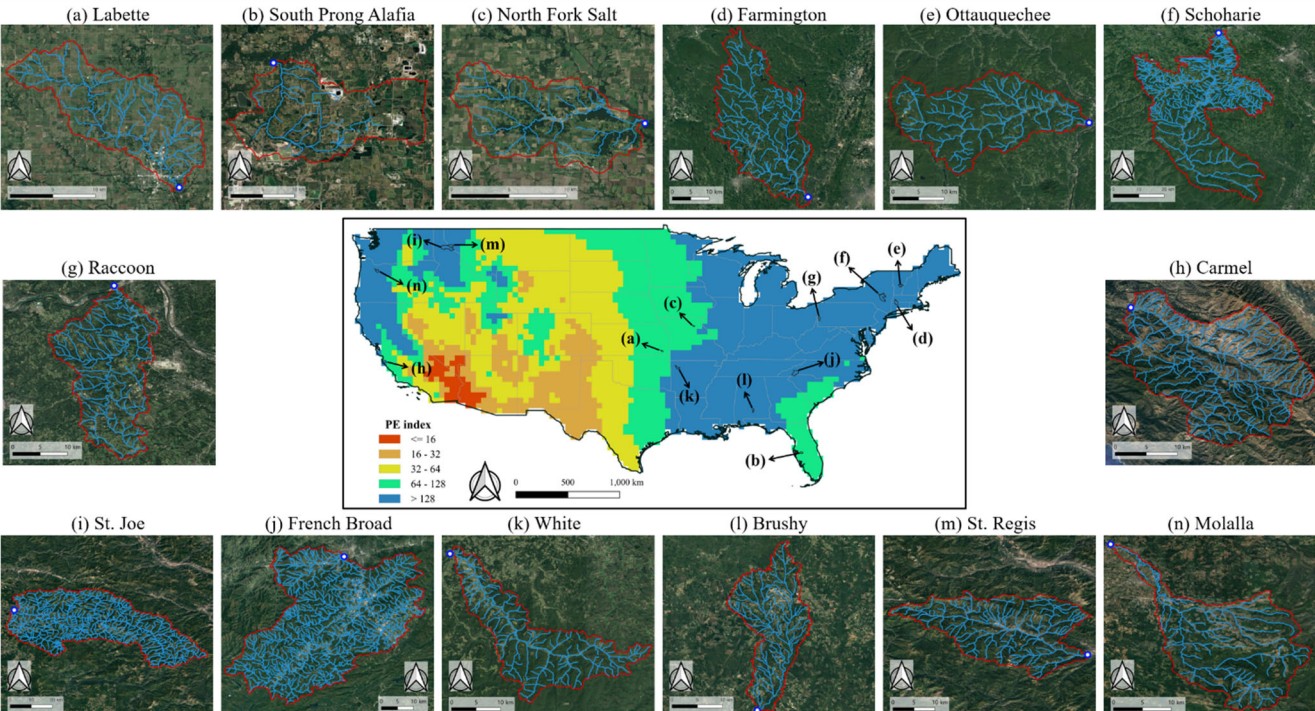

**Figure 2.** Structure and location of 14 river networks investigated in this study. The central map displays their geographic locations in the contiguous US overlaid with the spatial PE index distribution. Layouts of individual river networks surround the map, labeled from (a) to (n), corresponding to the order in Table 1. A circle mark in each figure represents the catchment outlet. The river network layouts (light blue lines) originate from NHDPlusV2. Satellite images in the background of the study areas were obtained from ©Google Earth.

To shape the structure of each river network at the grid domain, we used the 1 arc-second raster data of flow direction and upslope area provided in the National Hydrography Dataset Plus Version 2 (NHDPlusV2) (McKay et al., 2012). In NHDPlusV2, the Deterministic 8 method (O'Callaghan and Mark, 1984) is used for flow direction assignment. The flow direction extraction algorithm is underpinned by the principles of maximizing energy dissipation in surface water flow and minimizing power in groundwater flow (Schiavo et al., 2022). The DEM was post-processed to discard depression or sink cell. Accordingly, upslope area was calculated for each cell. For detailed calculation steps and process, readers may refer to the

user guide of NHDPlusV2. To extract river networks resembling individual blue-lines most, we referred to the source areas recorded in the NHDPlusV2. In NHDPlusV2, a channel forming area $A_o{}^*$ is given for stream channels at the most upstream points of individual flow paths in each river network. This is very detailed information, while $A_o$ in our notion is a single value which represents the entire network. We draw probability distribution of $A_o{}^*$ for each catchment (**Fig. S1** in the Supporting Information, SI) and $A_o$ was determined as the median (**Table 1**). Horton-Strahler ordering was assigned on the pruned river networks.

To investigate any impact of climatic forcing on Eq. (2), we analyzed the PE index (Thornthwaite, 1931), which is defined as the sum of the ratio of mean monthly precipitation to mean monthly potential evaporation (Wang and Wu, 2013). Note that a higher PE index indicates more moisture available for plant growth. We utilized precipitation and potential evapotranspiration data from the Climatic Research Unit Time-Series (CRU TS) on high-resolution 0.5-by-0.5 degree grids at a global scale (CRU TS v. 4.06 in Harris et al. (2022)) for the 50-year period from 1970 to 2019. The CRU dataset is compiled from a comprehensive collection of observations at weather stations.

Drawing the exceedance probability distribution of upstream area, i.e., $P(A \geq \delta)$, for a real catchment in log-log scale, three segments are often characterized: curved-head, straight-trunk, and truncated-tail. The power-law (Eq. (7)) holds for the straight trunk which indicates channels. The head reflects hillslope (Moglen and Bras, 1995; Maritan et al., 1996). As $A$ approaches $A_T$, the probability rapidly drops because the size of a network is finite (Rodríguez-Iturbe et al., 1992a; Moglen et al., 1998; Perera and Willgoose, 1998). To combine the channel part and the truncated tail in the distribution function, the exponentially tempered power function was adopted (Aban et al., 2006; Rinaldo et al., 2014) as

$$P(A \geq \delta) = c_d \delta^{-\varepsilon} \exp(-k_d \delta), \text{ for } \delta > A_o \tag{17}$$

where $c_d$ is a constant and $k_d$ is the tempering parameter. As $k_d$ approaches zero, the function represents abrupt truncation. Similarly, we proposed an exponentially truncated power function for $\rho_a$, as a general form of Eq. (2), as

$$\rho_a = c_p A_p{}^{-\eta} \exp(-k_p A_p), \text{ for } A_p > A_o \tag{18}$$

where $c_p$ is a constant and $k_p$ is the tempering parameter. To estimate the best-fitting parameters, we employed Matlab's *nlinfit* function which is designed for nonlinear regression for a given dataset. The objective of the function is to minimize the sum of the squares of the residuals for a defined nonlinear model. The estimated range for a parameter was calculated with 95% confidence intervals.

## 3.2 Results and discussion

All studied networks well follow the power-law Eq. (1) (**Fig. S2 in SI**). The range of estimated Hack's exponent $h$ is 0.55±0.03 (mean ± standard deviation) with $R^2 > 0.95$ (**Table 1**), which is within the typical range shown in earlier studies (Hack, 1957). The laws of stream number, length, drainage area, and eigenarea (Eq. (6)) are satisfied for all study networks with $R^2 > 0.85$ (**Figs. S3 – S4 in SI**). The resultant Horton ratios range as $R_B = 4.2 \pm 0.5$, $R_L = 2.3 \pm 0.3$, and $R_A = 4.6 \pm 0.7$ (**Table 1**), which are within typical ranges (Horton, 1945; Schumm, 1956; Smart, 1972). Further, $R_E = 2.2 \pm 0.3$, supporting the argument $R_E \approx R_L$ (Yang and Paik, 2017). These imply that our study networks hold statistically robust self-similar features.

In the exceedance probability distributions of upstream area, three segments of curved-head, straight-trunk, and truncated-tail are clearly characterized for all study catchments (**Fig. S5a in SI**). The visual interpretation is well demonstrated by the results of parameters fitted through Eq. (17) (mean squared error values $< 2 \times 10^{-8}$). The tempering parameter $k_d$ values are very small for all river networks, indicating an abrupt truncation in the tail part (**Table 1** and **Fig. S5b in SI**). The power-law exponent $\varepsilon$ ranges as $0.45 \pm 0.02$ (**Table 1**), which agrees with the range reported in earlier studies (e.g., Rodríguez-Iturbe et al., 1992a). $\varepsilon$ values estimated in our study networks satisfy the coupled relation with Hack's exponent $h$, resulting in $\varepsilon + h = 1.00 \pm 0.03$.

The $\rho_a$–$A_p$ relationship is plotted over all possible value of $A_p$ from the area of a single DEM cell (~900 m$^2$) to $A_T$. The plot closely resembles the $P(A \geq \delta)$ distribution, exhibiting the curved-head, straight-trunk, and truncated-tail (**Fig. 3a**). It is noteworthy that $A_o$ defined as the median of a given $A_o^*$ distribution aligns with the straight-trunk section for all studied rivers

(refer to **Table 1** for specific $A_o$ values). Notably, the three sections can be visually distinguished as two zones, i.e., Zone 1 illustrating the hillslope extent, and Zone 2 indicating the other two parts. Note that each catchment has its unique threshold for distinguishing between Zone 1 and Zone 2. The separation line drawn in Fig. 3a merely serves as a visual aid, ensuring efficiency in representing all studied catchments. Interestingly, the visually extracted $A_p$ value for the separation line closely approximates the minimum of all channel forming areas provided in NHDPlusV2.

In Zone 2, Eq. (18) satisfies quantitative description of the $\rho_a$–$A_p$ relationship for all study rivers (mean squared error values < 10$^{-3}$). The fitted tempering parameter $k_p$ is nearly zero, corroborating the extremely sharp cut-off in the tail of a distribution (**Fig. 3b** and **Table 1**). The power-law exponent $\eta$ ranges as 0.45 ± 0.04 (**Table 1**), which is close to but slightly smaller than the ranges of 0.48 ± 0.04 reported in Moglen et al. (1998) for 7 catchments with the median size of 30 km$^2$, and 0.47 ± 0.12 in Prancevic and Kirchner (2019) for 17 small mountainous catchments with the median size of 1.1 km$^2$. Integrating these earlier

empirical outcomes and results from this study, we can conclude that mostly $\eta < 0.5$. Further exploration linked to this dimensional inconsistency and fractal dimensions is given in the next section. We also investigated the functional distribution corresponding to hillslope, i.e., Zone 1. In our attempts, power-law function formatted as Eq. (2) seems applicable (**Fig. S6 in SI**). This is aligned with the findings of previous studies (Raff et al., 2004; Gangodagamage et al., 2011; Seybold et al., 2018). While hillslope area is outside of the scope of this study, this topic is worthy to be further investigated in subsequent research.

For every study network, the fitted $\eta$ value is very close to its $\varepsilon$ value (difference in % = 0.47 ± 0.30), which supports our theoretical derivation of $\varepsilon = \eta$ in Sect. 2.2. This means that the scaling exponent $\eta$ also has intimate relation with $h$ to be $\eta + h \sim 1$. In addition, the entire shapes of the two distributions are almost identical given $\varepsilon \approx \eta$ as well as $k_d \approx k_p$. The findings suggest that known physical meaning of $\varepsilon$ can provide insights into what $\eta$ physically stands for. By investigating the full range of binary trees from totally random to completely deterministic, Paik and Kumar (2007) highlighted that $\varepsilon$ represents how compact the

hierarchy of a given binary network is. Since they deal with tree topology, $\varepsilon$ can be more explicitly expressed as 'compactness of topological hierarchy.' In the consistent context, 'compactness of geometric hierarchy' can be symbolized by $\eta$ that is dependent on the concrete term of stream length.

Interestingly, the scaling exponent $\eta$ tends to be negatively related with the PE index (**Fig. 3c** and **Table 1**). In the mathematical aspect of the $\rho_a$–$A_p$ relationship, the decreasing linear regression model indicates that the total length of a river network ($L_T$)

formed in a catchment with higher PE index changes less sensitively when varying the pruning area ($A_p$). In the physical perspective, this finding suggests that a river network with a lower degree in the compactness of geometric hierarchy is likely to form in a landscape with greater availability of moisture for vegetation. The phenomenon is also hydrologically reasonable because surface water bodies, such as river networks, are naturally more pronounced in areas with an ample amount of groundwater, which is the dominant fraction of water resources used for vegetation survival (Mutzner et al., 2016; Zimmer

and McGlynn, 2017; Durighetto et al., 2022). Despite the plausible reasoning, we acknowledge the need for thorough follow-up research to explicitly demonstrate the joint contributions of climate and topography to $\eta$.

In contrast, our results reveal no significant distinction in $\eta$ values across the examined range of CSI. This suggests that, within the scope of this study, the relationship between $\rho_a$ and $A_p$ is not proportionally influenced by natural and anthropogenic pressures on the capacity of river reaches to flow freely. Future research covering a wider range of CSI than this study is expected

to provide a deeper understanding of how such forcing on free-flow river connectivity affects $\eta$.

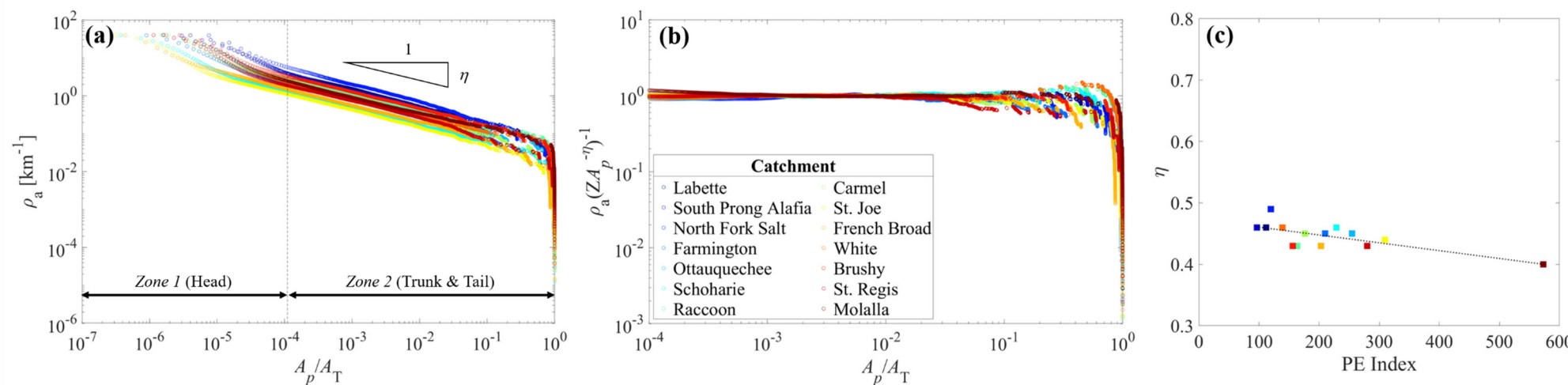

**Figure 3.** Analyses between the apparent drainage density $\rho_a$ and the pruning area $A_p$ for 14 studied catchments. Color-codes for each catchment are maintained consistently across all three panels presented herein. (a) Variation of $\rho_a$ with $A_p$ normalized by $A_T$ in a log-log scale. The dashed line differentiates Zone 1, which includes the curved-head part, from Zone 2, that encompasses both the straight-trunk and the truncated-tail segments. This line was visually extracted to ensure an efficient presentation, serving as a representative for all catchments. (b) Normalized $A_p$–$\rho_a$ distribution by individual power-law $\eta$ exponents. The x-axis range corresponds to Zone 2 in Fig. 3a. (c) Relationship between the scaling exponent $\eta$ and the PE index. The dotted line represents the linear regression fitted as $\eta = 0.47 - 1.28 \times 10^{-4}$ PE index, which is statistically significant ($p < 0.05$, $R^2 > 0.6$).

**Table 1.** Topographic characteristics of the 14 river networks analyzed in this study

| Catchment | State | Climate [1] (%) | CSI [2] | PE index | Final stream-order $\Omega$ | Total area $A_T$ (km²) | Source area $A_o$ (km²) | Horton ratios | | | | Hack's exponent $h$ | Area-exceedance probability distribution | | Apparent drainage density-pruning area relationship | | Fractal dimension | |
|---|---|---|---|---|---|---|---|---|---|---|---|---|---|---|---|---|---|---|
| | | | | | | | | $R_B$ | $R_L$ | $R_A$ | $R_E$ | | $\varepsilon$ | $k_d$ ($10^{-4}$) | $\eta$ | $k_p$ ($10^{-4}$) | $D_s$ | $D_b$ |
| Labette Creek* | KS | Dfa (100) | 58 | 112 | 5 | 222 | 0.21 | 4.2 | 2.3 | 4.8 | 2.1 | 0.60 | 0.46 | 20 | 0.46 | 22 | 1.0 | 1.8 |
| South Prong Alafia River* | FL | Cfa (100) | 65 | 96 | 5 | 350 | 0.32 | 4.1 | 2.3 | 4.4 | 2.2 | 0.52 | 0.47 | 27 | 0.46 | 28 | 1.1 | 1.7 |
| North Fork Salt River* | MO | Dfa (100) | 66 | 120 | 5 | 126 | 0.23 | 3.4 | 1.8 | 3.8 | 1.6 | 0.51 | 0.50 | 120 | 0.49 | 110 | 1.0 | 2.0 |
| Farmington | CT | Cfa, Cfb, Dfb (42, 42, 16) | 87 | 210 | 6 | 979 | 0.35 | 3.8 | 2.0 | 4.0 | 2.0 | 0.50 | 0.45 | 25 | 0.45 | 24 | 1.0 | 1.9 |
| Ottauquechee | VT | Dfb (100) | 94 | 255 | 6 | 572 | 0.55 | 3.1 | 1.8 | 3.0 | 1.8 | 0.53 | 0.45 | 11 | 0.45 | 11 | 1.1 | 1.9 |
| Schoharie | NY | Dfb, Dfa (99, 1) | 94 | 229 | 6 | 2,408 | 0.34 | 4.3 | 2.3 | 4.8 | 2.2 | 0.56 | 0.46 | 0.4 | 0.46 | 0.5 | 1.0 | 1.8 |
| Raccoon | PA | Dfa (100) | 96 | 164 | 5 | 476 | 0.20 | 5.0 | 3.0 | 5.4 | 2.8 | 0.58 | 0.43 | 3.3 | 0.43 | 3.6 | 1.3 | 1.4 |
| Carmel | CA | Csb, Csa (99, 1) | 96 | 176 | 6 | 593 | 0.13 | 4.1 | 2.2 | 4.5 | 2.2 | 0.53 | 0.45 | 24 | 0.45 | 26 | 1.0 | 1.8 |
| St. Joe | ID | Dsb, Dsc (86, 14) | 100 | 310 | 7 | 2,834 | 0.32 | 4.2 | 2.2 | 4.0 | 2.1 | 0.58 | 0.44 | 5.1 | 0.44 | 5.2 | 1.1 | 1.8 |
| French Broad | NC | Cfa, Cfb, Dfb (42, 42, 16) | 100 | 203 | 6 | 2,074 | 0.20 | 4.8 | 2.6 | 5.3 | 2.3 | 0.59 | 0.43 | 6.6 | 0.43 | 6.8 | 1.2 | 1.6 |
| White River | AR | Cfa (100) | 100 | 139 | 5 | 503 | 0.24 | 5.0 | 2.7 | 5.1 | 2.6 | 0.56 | 0.46 | 2.6 | 0.46 | 2.8 | 1.2 | 1.6 |
| Brushy | AL | Cfa (100) | 100 | 156 | 5 | 322 | 0.14 | 4.1 | 2.4 | 4.5 | 2.4 | 0.55 | 0.43 | 22 | 0.43 | 23 | 1.2 | 1.6 |
| St. Regis River | MT | Dsb, Dsc, Dfb, Dfc (54, 39, 5, 2) | 100 | 280 | 5 | 796 | 0.35 | 4.7 | 2.5 | 5.3 | 2.4 | 0.52 | 0.43 | 52 | 0.43 | 50 | 1.1 | 1.7 |
| Molalla River | OR | Csb, Csc, Dsb (90, 2, 8) | 100 | 573 | 5 | 569 | 0.47 | 4.1 | 2.5 | 4.8 | 2.2 | 0.58 | 0.40 | 7.1 | 0.40 | 7.4 | 1.2 | 1.6 |

**Note:** *Catchment name was referred from the Open Street Map as a creek or stream name at the outlet. (1) Climate zone was based on the Köppen climate classification scheme. (2) The reported Connectivity Status Index CSI was weighted by stream lengths for a given CSI.

## 4 Interpretation of Dimensional Inconsistency in $\eta$

It is worthwhile to investigate $\eta$ from dimensional perspective. Although $\eta = 0.5$ is anticipated for dimensional consistency (Tarboton et al., 1991), observed values are smaller than this in every network (see **Table 1**). As stated earlier, an analogous issue resides in Eq. (1): $h$ is expected to be 0.5 but observed values are mostly greater. This inconsistency was relaxed by introducing the fractal dimension of a stream as $D_s=2h$ (Mandelbrot, 1977), which was based on the assumption that the shapes of catchments are self-similar in a downstream direction (Feder, 1988; Rigon et al., 1996). For a stream reach, the fractal nature stems from stream sinuosity. Considering the typical range of $h$, $D_s$ is greater than unity, i.e., exceeding the dimension of a line, and mostly between 1 and 1.4 (Rosso et al., 1991). Motivated by this, we hypothesized that the deviation of the observed $\eta$ values from 0.5 implies the presence of non-integer fractal dimension of the topography.

We sought for the expression of $\eta$ as a function of fractal dimension, like $h = D_s/2$. As $\eta = \varepsilon = 1 - h$, from $h = D_s/2$ it is clear that

$$\eta = 1 - D_s/2. \tag{19}$$

We found that $\eta$ values estimated from Eq. (19) well agrees with observed values. However, above relationship becomes deceptive as Eq. (19) is identical to $\varepsilon + h = 1$ if $D_s=2h$ is applied. To resolve this issue, an independent relationship for $D_s$ should be introduced. We can employ the expression of $D_s$ from Horton ratios (Rosso et al., 1991) as

$$D_s = \max(1, 2\ln R_L/\ln R_A). \tag{20}$$

Two extreme values of $D_s$, i.e., 1 (a line with no sinuosity) and 2 (full sinuosity of streams filling a plane), correspond to cases of $R_A = R_L^2$ and $R_A = R_L$, respectively. Our 14 study networks show the $D_s$ range of $1.10 \pm 0.10$ (**Table 1**). Substituting Eq. (20) into Eq. (19) gives

$$\eta = 1 - \ln R_L/\ln R_A. \tag{21}$$

While $D_s$ represents the fractal dimension originated from the sinuous fractal stream (single corridor), there is another fractal nature stemming from the network organization of stream branches. Denoting the fractal dimension covering the latter feature as $D_b$, La Barbera and Roth (1994) derived an expression of $\varepsilon$ as a function of two fractal dimensions $D_s$ and $D_b$. As $\eta = \varepsilon$, we can use their derivation as

$$\eta = \varepsilon = D_s(D_b - 1)/2. \tag{22}$$

For $D_b$, we refer to the equation of La Barbera and Rosso (1989) as

$$D_b = \min(2, \ln R_B/\ln R_L). \tag{23}$$

According to Eq. (23), the lower and upper limits in $D_b$ (1 and 2) correspond to the cases of $R_B = R_L$ and $R_B = R_L^2$, respectively. Considering the typical ranges of $R_B$ and $R_L$ found in river networks, $D_b$ is mostly between 1.5 and 2 (La Barbera and Rosso, 1989; Rosso et al., 1991), and our study networks present $D_b$ ranging $1.73 \pm 0.16$ (**Table 1**). Substituting Eqs. (20) and (23) into (22) yields

$$\eta = \ln(R_B/R_L)/\ln R_A. \tag{24}$$

In that both $D_s$ and $D_b$ are considered, Eq. (24) is regarded as more comprehensive than Eq. (21). Indeed, Eq. (21) can be considered as a special form of Eq. (24) when $R_B = R_A$. As stated, empirical findings suggest $R_B \approx R_A$, but calculated $\eta$ can be sensitive to their differences. For $R_B < R_A$, which are found in most of our study networks (**Table 1**), Eq. (24) gives smaller value for $\eta$ than Eq. (21).

Besides Eq. (24), we can suggest another relationship which is from a very different perspective. Examining analyzed results, we found $\eta = \alpha D_b$, the linear tendency. Further, the coefficient is fairly invariant as $\alpha = 0.26 \pm 0.01$, from our 14 networks, which is very close to 1/4. Interestingly, this is similar to the quarter-power scaling laws widely found in self-similar biological

systems, such as the Kleiber's law (Kleiber, 1932; Ballesteros et al., 2018). Motivated by this finding and inspired by the simple expression of $h = D_s/2$, we suggest

$\eta = D_b/4 = (\ln R_B / \ln R_L)/4.$                                                          (25)

For all studied river networks, $\eta$ values estimated from Eqs. (24) and (25) have a high correlation coefficient of 0.95. Nonetheless, the two mathematical expressions for $\eta$ result a contrasting trend when compared with observed $\eta$ values from the $\rho_a$–$A_p$ relationship (**Fig. 4**). Eq. (24) yields greater deviations from observations, and mostly under-estimates $\eta$ values. It is interesting that the simple Eq. (25) is well supported by analysis results, with the estimated $\eta$ mean of 0.44 under merely ~6 %

difference from the observed $\eta$, which is around half of that calculated for Eq. (24). The inter-networks variability of the estimated $\eta$ for each equation is fairly similar to that of the observed values (standard deviation = 0.06 and 0.04 for Eqs. (24) and (25), respectively).

We perceive the poor performance of Eq. (24) as the consequence of weak assumptions which form the basis of theoretical derivations of Eqs. (20) and (23), i.e., Horton's laws hold precisely at all scales of a unit length to measure (La Barbera and

Rosso, 1989; Rosso et al., 1991). Indeed, this assumption is too ideal to be satisfied in real river networks, as corroborated in the non-perfect straight fits when estimating Horton's ratios of our studied networks (**Figs. S3- S4 in SI**). For $D_s$, the stream sinuosity cannot be directly analyzed with our DEM analysis due to limited resolution, and so large uncertainty is embedded. As a result, $D_s$ values estimated from Eq. (20) (shown in **Table 1**) differ from $D_s=2h$ with $h$ in **Table 1** (Mandelbrot, 1977). About $D_b$, Phillips (1993) who studied very small catchments in the Southern Appalachians in the USA also demonstrates that

satisfying the assumption is necessary to employ Eq. (23).

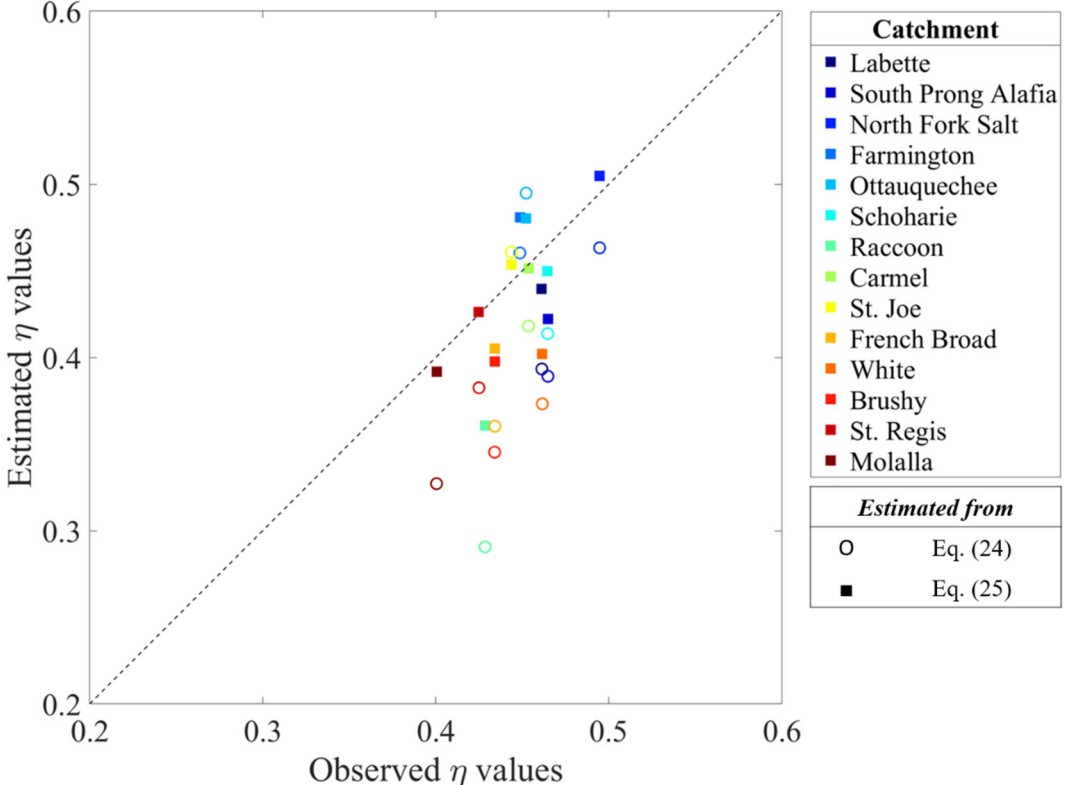

**Figure 4.** Comparison of $\eta$ value observed from the $\rho_a$–$A_p$ relationship (Eq. (18)), with $\eta$ values estimated as the functions of the fractal dimensions expressed as the Horton ratios. Results of Eqs. (24) and (25) are presented as hollow-circle and filled-square markers, respectively. Color-codes for our studied river networks are the same as indicated in Fig. 3.

As shown in Fig. 4, estimated/observed $\eta$ values are less than 0.5. This can be understood in three perspectives. First, taking Eq. (25), 0.5 becomes the upper limit of $\eta$, given the physical range of $1 \leq D_b \leq 2$. Second, the finding of $\eta < 0.5$ can also be understood from earlier studies on $\varepsilon$, given $\eta = \varepsilon$. In earlier studies about Eq. (7), $\varepsilon < 0.5$ is reported for most river networks (Rodríguez-Iturbe et al., 1992a; Crave and Davy, 1997). Although no attention has been given to the dimensional consistency in Eq. (7), in theory, random critical trees should follow $\varepsilon \approx 0.5$ (Harris, 1963). Paik and Kumar (2007) investigated trees, ranging from purely deterministic to completely random, and according to observed $\varepsilon$ values, river network organization is based on self-repetitive trees with some randomness in connectivity structure. In their follow-up study, Paik and Kumar (2011) dealt with more scaling laws of river networks to investigate the roles of the connectivity structures in tree organizations. Particularly for Hack's law analysis, they corroborated that partially random trees grounded on deterministic self-repetitive trees only exhibited the Hack's exponent $h$ within the range found from river networks.

Lastly, $\eta < 0.5$ can be explored from plausible optimality in the network formation. To explain physical mechanisms resulting the connectivity pattern of treelike river structures, various optimality hypotheses have been proposed, such as minimizing total energy expenditure (Rodríguez-Iturbe et al., 1992b; Rinaldo et al., 2006), total stream power (Chang, 1979), and total energy dissipation rate (Yang and Song, 1979), as summarized in (Paik and Kumar, 2010). Although debates on the physical mechanisms are still ongoing (Paik, 2012), the typical hypotheses share the underlying principle: ***direct connectivity from individual elements to a common outlet is maximized while total length of flow paths is minimized, in turn efficient flow connection under a given space***. It is noteworthy that optimal channel networks, which were created towards achieving the minimum total energy expenditure, showed the satisfactions of Hack's law with $h \sim 0.6$ (Ijjasz-Vasquez et al., 1993) and the area-exceedance probability distribution with $\varepsilon \sim 0.44$ (Bizzi et al., 2018; Carraro et al., 2020). The results suggest that the minimization of total energy expenditure needs to be considered not as a necessary condition but a sufficient condition. The notion of optimality resides in the quarter-power scaling laws which is linked to Eq. (25). West et al. (1997) suggested "an idealized zeroth-order theory" to explain the emergence of the quarter-power scaling laws in biological systems, based on three essential and generic properties of networks in organisms: (1) space filling to serve sufficient resources to everywhere in a system, (2) invariant size and characteristics of terminal units, and (3) optimized designs to minimize energy loss. According to their theory (West et al., 1999; West, 2017), the ubiquitous number 'four' in the scaling exponent indicates the total number of domains that all metabolic mechanisms are operated through optimized space-filling branching networks, thereby as a sum of the normal three domains representing three-dimensional appearance, and the additional one domain revealing fractal dimension feature. Indeed, it is broadly recognized that river network is an excellent analogue of biological networks in living organisms (Banavar et al., 1999). It implies that the interpretation for the number 'four' in the quarter-power scaling laws in biology may help to obtain a mechanism-based insight on the role of denominator 'four' in Eq. (25) for river networks of which fractal structures have been explained by optimality hypotheses.

## 5 Summary and Conclusions

Thorough investigations on the power-law relationship between the apparent drainage density $\rho_a$ and the pruning area $A_p$ with the exponent of $\eta$ were conducted. We unraveled the meanings of $\eta$ with dimensional inconsistency in diverse aspects. We analytically demonstrated that $\eta$ is equivalent to the fractal scaling exponent $\varepsilon$ in the area-exceedance probability distribution, based on a hypothetical network following the Hortonian tree framework. This pinpointed the coupled relationship between $\eta$ and Hack's exponent $h$ that is also deviated from the dimensional consistency, i.e., $(\eta = \varepsilon) + h = 1$.

Our arguments are well supported by evidence from many real river networks, covering wide ranges of climate condition and free-flow connectivity level over the contiguous United States, analyzed with NHDPlusV2 dataset. The $\rho_a$–$A_p$ relationships

for all studied catchments clearly exhibit curved-head, straight-trunk, and truncated-tail parts, which is identical shape as the area-exceedance probability distributions. Our findings highlighted that the empirical analyses results are in good agreement with the analytically found ones. It suggested that two scaling exponents $\eta$ and $\varepsilon$ are fundamentally identical but conceptually distinguishable, since geometric and topological attributes are inherent in the calculation procedure for $\eta$ and $\varepsilon$, respectively. Hence, we enabled to define physical meaning of $\eta$ as 'compactness of geometric hierarchy.'

Given the scaling exponent $\eta$ values for the studied catchments, we identified that they were negatively related with climate condition which represented as the precipitation effectiveness index, while not with free-flow connectivity level. The former finding was supported by not only physical aspect on the hierarchy of river network structure, but also hydrological mechanisms on the interaction between vegetation and the availability of surface water and groundwater. The latter finding implied that the exponent $\eta$ might not be linearly controlled by pressures on the capacity of river reaches to flow freely. Both findings provide compelling topics for follow-up research to deeply understand how climate and topography jointly contribute to $\eta$ and how forcing on free-flow connectivity affects $\eta$, respectively.

We further examined the physical implication of $\eta$ based on non-integer fractal dimensions. Such effort was elaborated as expressing $\eta$ as the functions of fractal dimensions on a single stream and the entire river organization, including the quarter-power scaling relationship. Despite the presence of inevitable uncertainty in quantifying fractal dimensions, the estimated $\eta$ values were likely aligned with the observed ones for all studied rivers. Given that, this study contributed to deeper understanding of the $\rho_a$–$A_p$ relationship. Our findings, further, lay the foundation of future studies on the interlinkage between fractal dimensions and indicators characterizing self-similar structures of river networks.

Overall, our study sites followed representative scaling laws of river networks, despite the differences in climate condition and connectivity level. In particular, our findings suggested that the interplay between $\varepsilon$ and $h$ for rivers is insensitive to the diverse conditions. It leads to a natural curiosity whether the diversity scope of the conditions was not sufficient or critical anthropogenic stressors were missing to uncover exceptional real river networks exhibiting the deviation from the well-known scaling properties. A follow-up study may need to resolve such curiosity with extended study sites at a global scale and additional descriptors for anthropogenic effects on river network structures and functions.

**Data availability**

This study did not use any new data to conduct the presented analyses. The National Hydrography Dataset Plus Version 2 for the contiguous U.S. are publicly available.

**Author contribution**

SY conceptualized this study and conducted initial analyses through her Master's thesis under KP's supervision. SY and KC performed the topographic analyses for the study networks, and interpreted them. SY and KP wrote the paper, and all co-authors reviewed and edited it. Funding was acquired by KP and SY.

**Competing interests**

The author declare that they have no conflict of interest.

## Acknowledgements

This work was supported by the National Research Foundation of Korea (NRF) grant funded by the Korea government (MIST) [RS-2023-00208991] and supported by Creative-Pioneering Researchers Program through Seoul National University [RS-0583-20230080]. We thank the Editor (Erwin Zehe) and two reviewers (Massimiliano Schiavo and Samuel Schroers) for their constructive comments.

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
