# Peer review of "Power-Law between the Apparent Drainage Density and the Pruning Area"

_Hydrology and Earth System Sciences, 2023_

## Referee Comment (RC2)

**Comments to the Authors:**

The work proposed by Dr. Yang, Dr. Choi, and Dr. Paik investigates the power-law (PL) relationship between the pruning area and the drainage density. The work's theoretical flavor does not lack in application, since it is applied to several rivers in the US as case studies. The overall impression of the work is good, whereas it can be improved in some parts and might benefit from further analysis. Major comments are itemized in the order of the manuscript's sections.

- **L. 20-25:** the literature-based explanation of the relationship between drainage density and climatic conditions of catchments can be elaborated a little bit more, e.g. providing further explanation about how $L_T$ increases at the decrease of $A_0$, and vice versa.

- **General comment:** you could provide a 2-panel Figure with sketched two kinds of basins for a prompt (visual) appraisal of the involved variables, in particular $A_0$ and $A_p$. They can be drawings of two different catchments you investigated, or just two exemplary (not real) ones, or again a sketch representing two catchments one upstream and one downstream of the same river.

- **Figure 1 and lines above:** the usage of 5 climatic regions in the US is forgotten in the rest of the manuscript. It seems that the networks, which are then investigated in their PL behaviors, are not related to the climatic region they are located anymore after Figure 1.
  Thus, there are two choices: 1) you can make the US Figure smaller and surround it (e.g. in enlargements) with as many Figures as the catchments you provide in Figure S1, and referring to climatic areas to generally frame these catchments and their overall climatic (hence precipitation? Please specify) conditions; 2) (I suggest this one) you can make a more impactful use of these climatic regions. For example, you can make a boxplot chart by classifying all the apparent drainage densities by climatic region. See, as an example, the classification of salinity values for depositional environments done by Schiavo et al. (2023) (Figure 5). If you choose to go down for keeping the subdivision in climatic zones of the US territory and therefore this influence on networks' structure, you should provide a classification of all the required exponents for all 5 climatic areas (see the following points).

- **Figure S1:** do all the DEM have the same spatial resolution? If yes, it's ok; if not, you should homogenize the meshes before employing any routing algorithm. Rigon et al. (1996) and Maritan et al. (2002) correctly underlined the multi-scaling problem when treating DEMs at different resolutions.

- **Figure 2a:** it is not clear to me which are the values of the 3 thresholds employed for discretizing the x-axis, i.e. the pruning area variable; they seem to be about 0.02, 0.5, and 3 $km^2$, respectively. How did you choose them? Could you report the numerical values and motivations (if any) you employed for these choices?

- **Figure 2a:** it would be nice to further discuss the PL relationship of each trait of the 3 portions you identified, i.e. values of apparent density for $A_p<0.02$, $0.02<A_p<0.3$, and $A_p>0.3$ (see e.g. D'odorico and Rigon, 2003). In other words, to offer results for the first, middle, and tail traits of the power-law relationship proposed in Figures 2a and 2b. This aspect would be very interesting to widen the investigation to the pruning area-apparent density relationships not only in the whole branch, but in each upstream, medium, and downstream trait. It also would provide a stronger confidence interval for estimating PL'exponents. Then, you'd come up with 4 exponents for each n=1,14 network, each exponent referring to a different portion of the network (like $\eta_1$, $\eta_2$, and $\eta_3$), and the 4th for the "whole" one ($\eta$, you already did this).

- **Figure 2b:** please comment on the impact of the threefold classification of PLs upon the variability of apparent density under varying pruning areas. These three kinds of exponents allow you to better investigate the behavior of the plot you give in this figure. Indeed, the 1st trait is constant, and the other two are sloping. Please comment adequately, also referring to each network portion's total area and branching structure.

- **Figure 3:** can you give the correlation coefficient of all those exponents obtained upon the (24) and those with the (25) to quantify the correlation discrepancies? Moreover, could you provide the other 3

panels of the same Figure by plotting η1, η2, and η3 in the same way (and with them correlation coefficients)?

- **Flow routing method:** the choice of the D8 flow method is not adequately supported. Indeed, it has been probably (implicitly?) chosen concerning the DInf or other ones because D8 guarantees the maximum energy (local) dissipation (e.g. in Schiavo et al. (2022)) by always following the steepest descent. Please clarify this point, eventually referring to Schiavo et al. (2022), for a complete thermodynamic framework (please elaborate on these concepts a little bit) of the processes you are investigating.

Once all these points are solved, I consider the paper interesting and innovative, and worthy of being published in such a prestigious journal. Best regards,

Massimiliano Schiavo

---

## Author Response (AR1)

**Point-by-point Reply to the Comments**

Dear Prof. Zehe (the editor),

We do appreciate your providing us with the revision decision for HESS-2022-237. The submitted revised manuscript incorporates nearly all the responses described in the Author Comments (AC), which was already uploaded to the interactive manuscript discussion page. These are included in this file with changes in content and corresponding line numbers reflected in the submitted revised manuscript.

In fact, while revising the manuscript, we realized that our proposed direction for revision and the corresponding statements addressing two comments (R2C5 and R2C6, where 'R' stands for Reviewer and 'C' for Comment) needed more careful consideration. Consequently, the revised manuscript inevitably contains content that differs what was originally written in the AC. These inevitable modifications were made to address two specific topics, each with its core rationale, as summarized below.

Topic 1 : The identification of hillslope, channel, and tail parts in the $\rho_a$–$A_p$ relationship

- In the submitted revised manuscript, the parts are distinguished into two zones, i.e., Zone 1 for the hillslope extent and Zone 2 for the other two parts (see Fig. 3a). The primary reason was to improve visual efficiency in presenting the distribution, ensuring alignment with the fitting coverage applied to the exponentially tempered power function.

Topic 2 : The examination of a power-law scaling in the hillslope extent

- In the submitted revised manuscript, we present our analysis and the corresponding findings on the scale-invariant behavior as a compelling topic for follow-up research (see Fig. S6 in SI). This approach was taken to prevent deviation from the study's primary novel contribution of this study: the power-law relation between the apparent drainage density and the pruning area for river networks, while also addressing the reviewers' constructive feedback on this topic.

Thank you very much for your and the reviewers' understanding regarding the inevitable modifications.

**Author Comment – Referee 1**

We thank Referee 1 for constructive comments. Our response (in blue) to each comment is listed below, with changes in content and corresponding line numbers reflected in the submitted revised manuscript.

**Comment 1:** You mention that the theoretical values of epsilon and h should be 0.5 but they seem to be below 0.5 (epsilon) and above 0.5 (h). You then mention that the theoretical values are related to dimensional consistency and that the found dimensional inconsistency is related to fractal dimension of topography. Although I understand your reasoning it is a bit difficult to follow. What is the fractal dimension of topography? I think it could help to try to underly these concepts additionally with physical meaning, e.g. a stream is sinusoidal because total loss of energy is minimized, similarly one could argue that a network tends to be compacted in hierarchy because.... It is noteworthy to include a sentence on how fractals and flow connectivity are interrelated and what this physically means (energy).

**Response 1:** Fractal dimensions of river networks vary between 1 and 2, as explained in L240-241 and L260 of the original manuscript. The positive non-integer dimension indicates that, in a planar view, a single stream in a given river network is more winding than a straight line, which has a dimension of 1, and simultaneously the entire layout of the network does not completely fill a given surface, which has a dimension of 2.

Regarding the comment to explain the fractal features of river networks based on their physical means, we fully agree with your idea and the associated reason. Particularly, we wish if we could clearly elucidate "which physical mechanism results in which fractal feature", like the example statement you provided. However, it must be acknowledged that many hypotheses have been proposed to unravel which physical processes induce river network structures to be fractal, resulting in persistent debates on the topic (refer to L302-306 of the original manuscript). Thereby, it is hardly to explain the causality based on a one-to-one approach. Nonetheless, to do our best within the scope of this study, we clarified the underlying mechanical principle out of proposed hypotheses so far (refer to L306-308 of the original manuscript).

Overall, to incorporate your constructive comment, **we improved the original L62 as follows (L66-68 in the revised manuscript).** It aims to provide additional introductory explanation on the fractal dimension in the context of this research:

"This has brought the introduction of the fractal dimension (Mandelbrot, 1977), whose values for river networks range between 1 and 2 (e.g., Feder, 1988) (further detailed explanations are provided in Sect. 4)."

**Comment 2:** Another minor comment I have regarding the range of application of the found power law. There seem to be different minimum pruning areas for the law to be applicable. How can this be justified, and could there be a companion power law for smaller pruning areas? If so or if not, why?

**Response 2:** You correctly understood that individual catchments analyzed in this study have distinct values of minimum pruning area which is eventually equivalent to a source area $A_o$. For a given catchment, $A_o$ is a constant determined as the median of channel forming areas extracted from the National Hydrography Dataset Plus Version 2 (NHDPlusV2). The applied $A_o$ values in this study can be justified because NHDPlusV2 represents the blue-lines of river networks spanning the contiguous US (refer to L174-177 of the original manuscript).

For our analysis, we selected the median of all channel forming areas by following this logic: the median represents 50% frequency of total cases of channel forming areas (i.e., half of total case). As a typical starting point, we wanted to comply with the criterion for at least half of all cases to define river networks. The threshold 50% can therefore be directly justified as a neutral choice to set spatially constant source area for a given catchment.

**Specific comments:**

**Comment 3:** L22: Is there some physical explanation for this?

**Response 3:** Yes. The precipitation effectiveness index (PE index) is expressed as the sum of the ratio of mean monthly precipitation to mean monthly evaporation over a year (Thornthwaite, 1931). Hence, the PE index indicates how much moisture is available for plant growth. Considering the context of this study, **we elaborated it in L22-24 and L204-206 of the revised manuscript as follows:**

"The spatial variation of $\rho$ among catchments is associated with their climates (Melton, 1957; Madduma Bandara, 1974; Wang and Wu, 2013), which can be represented by measures such as the precipitation effectiveness (PE) index (Thornthwaite, 1931)."

"..we analyzed the PE index (Thornthwaite, 1931), which is defined as the sum of the ratio of mean monthly precipitation to mean monthly potential evaporation (Wang and Wu, 2013). Note that a higher PE index indicates more moisture available for plant growth."

**Comment 4:** L26: Isn´t this the question? How can the topography be "given" if these apparent power laws can be found? There seems to be some organization, so I would suggest to mention this here.

**Response 4:** Thank you for pointing it out. It is general phenomenon that the drainage density varies with the source area for any river network. **We corrected it in L28-29 of the revised manuscript as follows:**

"On another note, the 'rate' at which LT (and so $\rho$) varies with Ao is likely determined by the shape of landscape or a given topography."

**Comment 5:** L47: some concepts might need more explanation, what is valley transmissivity in this context?

**Response 5:** Thank you for pointing it out. **We added the physical expression and dimension of valley transmissivity in L50-52 of the revised manuscript as follows:**

"$\gamma$ is the exponent of a hypothetical power function between A and valley transmissivity T (the product of subsurface cross-sectional area and conductivity, which in turn is expressed in units of cubic length per time (Prancevic and Kirchner, 2019)."

**Comment 6:** L85: I think some simpler variable for cumulative length might be better for readability.

**Response 6:**  Thank you for the advice. **We replaced the original notation '$\Theta_w$' of the cumulative mean length with a new notion in L94-95 of the revised manuscript as follows:**

"… the cumulative mean length $\Gamma_\omega$ was proposed to match the definition of area as"

**Comment 7:** L106: Some additional explanation for this equation would help the reader.

**Response 7:**  Thank you for the suggestion. To resolve your concern, we will add supporting explanation on the concept of Eq. (7) in the revised manuscript.

**Comment 8:** L108: What is meant by such a trade-off?

**Response 8:**  In the context, a trade-off means that mechanisms and processes forming river networks harmonize with each other under limited space, manifesting a relation between two scaling exponents, h and ε, as h + ε = 1. As we stated in the manuscript, the two scaling exponents, h and ε, are within the respective narrow ranges of h = 0.5 ~ 0.7 and ε = 0.4 ~ 0.46. Furthermore, the interdependence between the two exponents is found to be h + ε = 1. Given that, the trade-off between h and ε can be understood by comparing two catchments (X and Y): if catchment X has a higher h value than catchment Y, then the ε value of X is smaller than that of Y, provided both exponents are within their respective narrow ranges.

To deliver the meaning of the trade-off more clearly, **the original L108 were edited in L120-122 of the revised manuscript, as follows:**

"h + ε = 1 (Maritan et al., 1996), which suggests a trade-off between the two exponents by balancing each other with their respective ranges to form the catchment boundary within a confined space."

**Comment 9:** L171: Is the analysis sensitive to the method used to derive flow direction?

**Response 9:** We expect no effect of flow direction algorithms on the analysis result. Our response is based on comprehensive references on diverse algorithms to extract flow directions. It is inevitable because no previous study on the $\rho_0$-$A_p$ relationship has been conducted by using the other method of extracting either single or multiple flow direction.

In general, when producing more accurate and smoother geometric patterns, algorithms to define multiple flow direction (i.e., flow of a cell drains into all downslope neighboring cells) are likely to be superior to single flow direction algorithm (i.e., drainage of a cell occurs in the steepest downslope direction) (Quinn et al., 1991; Costa-Cabral and Burges, 1994; Pan et al., 2004). However, hydrologic responses of a given catchment are almost no different between single and multiple flow directions (Wolock and McCabe Jr., 1995). This suggests that the accuracy of flow paths at the local scale is compromised as the scope expands to encompass the whole catchment.

Furthermore, we do strongly anticipate the insensitivity of scaling features to a way to define flow direction, based on the study of Paik (2011). The reference demonstrates that power law relationships in Hack's law and the area-exceedance probability distribution are manifested in river networks extracted by the other single flow direction algorithm for determining flow direction – the Global Deterministic 8 method newly developed by Paik (2008).

**Comment 10:** L175: Reach or stream?

**Response 10:** Thank you for pointing it out. To further clarify, **the original L175 was replaced by the following sentence in L199-200 of the revised manuscript:**

"In NHDPlusV2, a channel forming area $A_o^*$ is given for stream channels at the most upstream points of individual flow paths in each river network."

**Comment 11:** L188: You mention that you applied a linear fit. This refers to the logarithms? As one can also see in Fig. 2a this fit is only valid for a certain interval of values, therefore how did you derive the lower cutoff value?

**Response 11:** It seems that you were confused by the part 'linfit' in Matlab's *nlinfit* function. The function is designed not for fitting a linear regression but for a nonlinear regression for a given dataset. Indeed, the *nlinfit* function aims to identify the best-fitting parameters by minimizing the sum of the squares of the residuals for a defined nonlinear model. In this study, we applied the *nlinfit* function to estimate the best parameters fitting Eqs. (17) and (18), respectively.

**To clarify it, we described additional explanation in L220-222 of the revised manuscript as follows:**

"To estimate the best-fitting parameters, we employed Matlab's *nlinfit* function which is designed for nonlinear regression for a given dataset. The objective of the function is to minimize the sum of the squares of the residuals for a defined nonlinear model."

**Comment 12:** L225: Fig. 2a: Different $A_0$ values have been found, what might the range between the minimum $A_0$ and the maximum $A_0$ be related to?

**Response 12:** Source area $A_0$ is intimately influenced by diverse conditions characterizing a catchment, such as hydrological, climatic, geomorphological, and geological conditions. Particularly, hydro-climatic conditions are significantly related to the source area and its corresponding drainage density. This relationship is well justified by the Abraham curve (Abrahams, 1984), which shows that drainage density of a river network decreases in arid regions and increases in humid regions. Thereby, follow-up research using the newly analyzed PE index for our studied catchments is expected to provide deeper understanding on how the variety of environmental conditions would affect the formation of river networks.

**Author Comment – Referee 2**

We thank the Referee 2 for constructive comments. Our response (in blue) to individual comments is listed below with changes in content and corresponding line numbers reflected in the submitted revised manuscript.

**Comment 1:** L. 20-25: the literature-based explanation of the relationship between drainage density and climatic conditions of catchments can be elaborated a little bit more, e.g. providing further explanation about how $L_T$ increases at the decrease of $A_0$, and vice versa.

**Response 1:** Thank you for your constructive comment. To clarify the intermediate mechanism, **we substituted the original L20-25 with the following sentences in L21-27 of the revised manuscript:**

"The spatial variation of $\rho$ among catchments is associated with their climates (Melton, 1957; Madduma Bandara, 1974; Wang and Wu, 2013), which can be represented by measures such as the precipitation effectiveness (PE) index (Thornthwaite, 1931). Also over time, $A_o$ and so $\rho$ of a given catchment dynamically vary. $A_o$ reduces as the catchment becomes wetter, water accumulates more readily in the soils of low-gradient areas, and saturated areas expand accordingly. This mechanism leads to the enlargement of the stream network (greater $L_T$). Conversely, when the catchment gets drier, $A_o$ increases, which in turn results in the contraction of the stream network (Godsey and Kirchner, 2014; Hooshyar et al., 2015; Durighetto et al., 2020)."

**Comment 2:** General comment: you could provide a 2-panel Figure with sketched two kinds of basins for a prompt (visual) appraisal of the involved variables, in particular $A_0$ and $A_p$. They can be drawings of two different catchments you investigated, or just two exemplary (not real) ones, or again a sketch representing two catchments one upstream and one downstream of the same river.

**Response 2:** Thank you very much for the constructive comment. In Sect. 2.2 of the revised manuscript, we added a new figure that conceptually explains the core variables studied in this work. **This was designated as Figure 1 in the revised manuscript.**

**Comment 3:** Figure 1 and lines above: the usage of 5 climatic regions in the US is forgotten in the rest of the manuscript. It seems that the networks, which are then investigated in their PL behaviors, are not related to the climatic region they are located anymore after Figure 1. Thus, there are two choices: 1) you can make the US Figure smaller and surround it (e.g. in enlargements) with as many Figures as the catchments you provide in Figure S1, and referring to climatic areas to generally frame these catchments and their overall climatic (hence precipitation? Please specify) conditions; 2) (I suggest this one) you can make a more impactful use of these climatic regions. For example, you can make a boxplot chart by classifying all the apparent drainage densities by climatic region. See, as an example, the classification of salinity values for

depositional environments done by Schiavo et al. (2023) (Figure 5). If you choose to go down for keeping the subdivision in climatic zones of the US territory and therefore this influence on networks' structure, you should provide a classification of all the required exponents for all 5 climatic areas (see the following points).

**Response 3:** Thank you very much for your considerate comment. After deep consideration, we have decided to incorporate the climatic feature merely as one of explanatory variables for the study areas. (i.e., the first option you suggested). The primary reason for this decision is that a climate-dependent classification of the analyzed power law exponents does not yield impactful findings. In other words, it demonstrates that river networks formed in different climate zones surprisingly reveal consistent attributes in describing fractal structures.

To provide more suitable guiding information at an earlier stage, we added a new figure that presents the layouts and locations of the studied14 river networks, by merging Figure S1 with adjusted Figure 1 from the original manuscript. **This new figure was designated as Figure 2 in the revised manuscript.**

**Comment 4:** Figure S1: do all the DEM have the same spatial resolution? If yes, it's ok; if not, you should homogenize the meshes before employing any routing algorithm. Rigon et al. (1996) and Maritan et al. (2002) correctly underlined the multi-scaling problem when treating DEMs at different resolutions.

**Response 4:** Yes. All DEMs analyzed have the same resolution.

**Comment 5:** Figure 2a: it is not clear to me which are the values of the 3 thresholds employed for discretizing the x-axis, i.e. the pruning area variable; they seem to be about 0.02, 0.5, and 3 km$^2$, respectively. How did you choose them? Could you report the numerical values and motivations (if any) you employed for these choices?

**Response 5:** We'd like to remind that only one threshold of $A_p$ was applied in the x-axis to characterize the $\rho_0$-$A_p$ relationship. The threshold $A_p$ value is defined as a source area $A_0$ for a given catchment. The value of $A_0$ is a constant determined as the median of channel forming areas extracted from the National Hydrography Dataset Plus Version 2 (NHDPlusV2). All relevant descriptions are given to Table 1, L174-177 and L206-207 of the original manuscript.

Despite the existing explanation in the manuscript, we fully anticipate that the original Figure 2a can confuse readers as you addressed those questions. **To resolve the expected issue, we revised it as new Figure 3a in the revised manuscript.**

**Comment 6:** Figure 2a: it would be nice to further discuss the PL relationship of each trait of the 3 portions you identified, i.e. values of apparent density for $A_p<0.02$, $0.02<A_p<0.3$, and $A_p>0.3$ (see e.g. D'odorico and Rigon, 2003). In other words, to offer results for the first, middle, and tail traits of the power-law relationship proposed in

Figures 2a and 2b. This aspect would be very interesting to widen the investigation to the pruning area-apparent density relationships not only in the whole branch, but in each upstream, medium, and downstream trait. It also would provide a stronger confidence interval for estimating PL'exponents. Then, you'd come up with 4 exponents for each n=1,14 network, each exponent referring to a different portion of the network (like $\eta 1$, $\eta 2$, and $\eta 3$), and the 4th for the "whole" one ($\eta$, you already did this).

**Response 6:** Linking to our Response 5, your Comment 6 is interpreted to inquire the existence of a power function even in the range of $A_p < A_0$. It is a very interesting and insightful topic to study further because the power function is highly likely to corroborate the self-similarity in the hillslope extent (e.g., Raff et al., 2004; Gangodagamage et al., 2011; Saybold et al., 2018).

Interestingly, the insight on the companion power-law had been also addressed during our work. It led us to apply a power function to the hillslope extent illustrated as Zone 1 in the new Fig. 3a. **We presented our analysis and the corresponding findings on the scale-invariant behavior as a compelling topic for follow-up research (see Fig. S6 in SI).**

**Comment 7:** Figure 2b: please comment on the impact of the threefold classification of PLs upon the variability of apparent density under varying pruning areas. These three kinds of exponents allow you to better investigate the behavior of the plot you give in this figure. Indeed, the 1st trait is constant, and the other two are sloping. Please comment adequately, also referring to each network portion's total area and branching structure.

**Response 7:** Thank you for the careful comment. In fact, the threshold classification is not useful in describing details of the original Figure 2b (i.e., the normalized $\rho_0$-$A_p$ distribution). Because it is aimed to clearly demonstrate the power law behavior captured in the trunk part of the original Figure 2a.

To clarify the issue, **we added the explanation in the caption of the new Figure 3 as follows:**

"The x-axis range corresponds to Zone 2 in Fig. 3a."

**Comment 8:** Figure 3: can you give the correlation coefficient of all those exponents obtained upon the (24) and those with the (25) to quantify the correlation discrepancies? Moreover, could you provide the other 3 panels of the same Figure by plotting $\eta 1$, $\eta 2$, and $\eta 3$ in the same way (and with them correlation coefficients)?

**Response 8:** Thank you for your constructive comment. **We added the following sentence in L326 of the revised manuscript,** to provide the correlation coefficient between $\eta$ values from Eqs. (24) and (25)

"For all studied river networks, $\eta$ values estimated from Eqs. (24) and (25) have a high correlation coefficient of 0.95."

Regarding the suggested extra panels, we recognize its importance and potential support when additional analyses will be carried out to identify the power-law features found in different extents of the $\rho_0$-$A_p$ distribution. As stated in our Response 6, the suggested aspect falls outside the scope of the current study after our careful consideration.

**Comment 9:** Flow routing method: the choice of the D8 flow method is not adequately supported. Indeed, it has been probably (implicitly?) chosen concerning the DInf or other ones because D8 guarantees the maximum energy (local) dissipation (e.g. in Schiavo et al. (2022)) by always following the steepest descent. Please clarify this point, eventually referring to Schiavo et al. (2022), for a complete thermodynamic framework (please elaborate on these concepts a little bit) of the processes you are investigating.

**Response 9:** Thank you for the valuable comment. Firstly, we'd like to note that the D8 method for allocating flow direction was selected not by us, but by the research team formulating the NHDplusV2 dataset. Unfortunately, any reasoning behind the choice of the D8 method is not justified in the NHDplusV2 User Guide (McKay et al., 2012). Given that, we believe that providing an exact answer to your comment is beyond the scope of our responsibility.

Nonetheless, here we'd like to response as best as we can based on our knowledge of flow direction extraction algorithms. In general, when producing more accurate and smoother geometric patterns, algorithms to define multiple flow direction (i.e., flow of a cell drains into all downslope neighboring cells) are likely to be superior to single flow direction algorithm (i.e., drainage of a cell occurs in the steepest downslope direction) (Quinn et al., 1991; Costa-Cabral and Burges, 1994; Pan et al., 2004). However, hydrologic responses of a given catchment are almost no different between single and multiple flow directions (Wolock and McCabe Jr., 1995). This suggests that the accuracy of flow paths at the local scale is compromised as the scope expands to encompass the whole catchment.

Furthermore, we do strongly anticipate the insensitivity of scaling features to a way to define flow direction, based on the study of Paik (2011). The reference demonstrates that power law relationships in Hack's law and the area-exceedance probability distribution are manifested in river networks extracted by the other single flow direction algorithm for determining flow direction – the Global Deterministic 8 method newly developed by Paik (2008).

**To clarify the suitability in the D8 method application, we advanced the original L171-172 as the following sentences in L193-196 of the revised manuscript :**

"In NHDPlusV2, the Deterministic 8 method (O'Callaghan and Mark, 1984) is used for flow direction assignment. The flow direction extraction algorithm is underpinned by the principles of maximizing energy dissipation in surface water flow and minimizing power in groundwater flow (Schiavo et al., 2022)."